# LSO-FastSLAM: A New Algorithm to Improve the Accuracy of Localization and Mapping for Rescue Robots

**DOI:** 10.3390/s22031297

**Published:** 2022-02-08

**Authors:** Daixian Zhu, Yinan Ma, Mingbo Wang, Jing Yang, Yichen Yin, Shulin Liu

**Affiliations:** 1College of Communication and Information Engineering, Xi’an University of Science and Technology, Xi’an 710054, China; crazyboymyn@163.com (Y.M.); jetyyc@163.com (Y.Y.); 2College of Energy Science and Engineering, Xi’an University of Science and Technology, Xi’an 710054, China; 3Xi’an Xiangteng Microelectronics Technology Co., Ltd., Xi’an 710054, China; crazygirlyj@163.com; 4College of Electrical and Control Engineering, Xi’an University of Science and Technology, Xi’an 710054, China; lsigma@163.com

**Keywords:** mine robot, simultaneous localization and mapping, FastSLAM, lion swarm optimization algorithm, particle weight, particle diversity

## Abstract

This paper improves the accuracy of a mine robot’s positioning and mapping for rapid rescue. Specifically, we improved the FastSLAM algorithm inspired by the lion swarm optimization method. Through the division of labor between different individuals in the lion swarm optimization algorithm, the optimized particle set distribution after importance sampling in the FastSLAM algorithm is realized. The particles are distributed in a high likelihood area, thereby solving the problem of particle weight degradation. Meanwhile, the diversity of particles is increased since the foraging methods between individuals in the lion swarm algorithm are different so that improving the accuracy of the robot’s positioning and mapping. The experimental results confirmed the improvement of the algorithm and the accuracy of the robot.

## 1. Introduction

Mine rescue robot plays a critical role in mine rescues [1,2,3]. When a mine accident occurs, it can replace the rescue workers to search and rescue so as to protect rescuers from the great amount of combustible gas and dust in the mine. In particular, due to the complex environment after the disaster, rescuers cannot enter the narrow space, so they need to use robots to search and rescue. Moreover, the robot can determine its own position through the map established by itself and guide the trapped people to leave the site after the disaster. Hence, an accurate map under the mine has great significance for fast rescue. However, when the mine accident occurs, the underground environment of the mine changes greatly, and the original underground map is not suitable for the current rescue environment. Although the related positioning system is equipped under the mine, the positioning equipment is damaged, and the precise positioning function cannot be realized when the mine accident occurs. Therefore, simultaneous localization and mapping (SLAM) of robots play a significant role in mine rescue.

The concept of SLAM is proposed to deal with the problem of simultaneous positioning and mapping of robots [4,5,6]. In an unknown environment, the robot combines its own sensors such as lidar and vision cameras to estimate its own pose and build an environment map [7,8]. The classical SLAM algorithm research is based on filtering theory, which includes Kalman filter-based EKF-SLAM [9] and particle filter-based FastSLAM algorithm [10]. 

Among them, EKF-SLAM can only handle Gaussian noise, and its processing speed is slow for non-Gaussian noise environment. Moreover, in the same scenario, its processing speed is slower than the FastSLAM algorithm.

However, there are some potential defects in the FastSLAM algorithm. One of the problems is particle weight degradation and particle diversity loss. This problem is caused by the adoption of Rao-Blackwellized filter in the FastSLAM algorithm, which will decrease the accuracy of the FastSLAM algorithm [11,12,13,14,15]. Another problem of the FastSLAM algorithm is that it requires a large number of particles to maintain the accuracy of the algorithm in a complex environment, which will increase the running time of the algorithm and reduce the efficiency of the robot. In addition, the accurancy of algorithm decreased once the number of particles is insufficient [16].

In order to overcome the drawbacks of the FastSLAM algorithm, researchers put forward different ideas for algorithm improvement. Kim, together with his coworkers, proposed the unscented FastSLAM (UFastSLAM) algorithm. Based on the FastSLAM algorithm, they used an unscented Kalman filter (UKF) instead of an extended Kalman filter (EKF) for robot pose estimation, which improved the filtering accuracy of the robot [17]. However, they did not solve the particle degradation problem. HeBo et al. solved the problem of particle weight degradation in the UFastSLAM algorithm through particle swarm optimization (PSO) algorithm, and tested it on their autonomous underwater vehicle (AUV), which verified the effectiveness and feasibility of the improved algorithm [18]. Different from the above methods, with the deepening of the research on the meta-heuristic algorithm [19,20,21,22,23], more and more researchers use the meta-heuristic algorithm to improve the FastSLAM algorithm [24,25,26,27,28]. 

Tian and his coworkers optimized the FastSLAM through the improved firefly algorithm to improve the robot positioning with the accuracy of mapping. Meanwhile, this method effectively solved the problem of particle degradation [29]. Tian et al. used the meta-priming algorithm to optimise the importance sampling of the FastSLAM algorithm, thus effectively solving the degradation of particle weights and loss of particle diversity, and improved the robot localization mapping accuracy. In contrast, some literature [25] focused on the optimization of the resampling process in the FastSLAM algorithm, and optimized the resampling process of the FastSLAM algorithm by the particle swarm algorithm to achieve the increased localization mapping accuracy of the robot. Although the metaheuristic algorithm has achieved impressive results in optimising the FastSLAM algorithm, the algorithm is prone to fall into local optimality due to the strong optimisation capability of the metaheuristic algorithm; in addition, using the metaheuristic algorithm to optimise the FastSLAM algorithm, the fusion of multiple algorithms will lead to a significant increase in the algorithm time complexity.

In order to solve the problem of low accuracy of mine rescue robot positioning and mapping, this paper uses the FastSLAM2.0 algorithm to decompose the robot positioning and mapping into a moving part and a conditional map part to reduce the sampling space. However, due to the degradation of particle weights and the loss of particle diversity in the FastSLAM2.0 algorithm, the accuracy of robot positioning and mapping is reduced. Based on reading a large number of the FastSLAM algorithm improvement documents, this article adopts a new meta-heuristic algorithm lion swarm optimization (LSO) algorithm [30], to optimize the FastSLAM2.0 algorithm. In our work, we optimized the particle distribution after importance sampling through the LSO algorithm and regarded the sample particle set as a lion group, and optimized the individuals’ movement law between the populations in the lion group algorithm. The sample particle distribution solves particle degradation. At the same time, in order to keep particle diversity, we uses genetic algorithm to replace the lioness movement process to increase the particle diversity, and finally achieve the degradation of the particle weight of the FastSLAM2.0 algorithm. This solution to the loss of particle diversity can improve the accuracy of robot positioning and mapping.

The specific contributions of this paper are as follows:This paper designs and implements a new scheme to optimise the FastSLAM algorithm by means of the Lion Swarm algorithm.In the process of optimizing the FastSLAM algorithm through the lion swarm algorithm, the distribution of the particle set after important sampling in the FastSLAM algorithm is achieved through the division of labour between different individuals in the lion swarm optimisation algorithm, so that the particles are distributed in the high likelihood region, and solving the particle weight degradation problem. In addition, to ensure particle diversity, a genetic algorithm is used instead of the lioness movement process in the lion swarm to further increase the particle diversity.In this paper, the innovative FastSLAM algorithm is applied to a rescue robot by optimizing the Lion Swarm algorithm, aiming to improve the localization and mapping accuracy of the rescue robot.

The rest of the paper is organized as follows. Section 2 introduces the FastSLAM briefly. Section 3 Introduces the principle of lion swarm optimization algorithm. Section 4 discusses the proposed LS0-FastSLAM in detail. Section 5 gives the simulation results and analyzes the performance of the LSO-FastSLAM in detail. Finally, Section 6 concludes the paper and provides some suggestions for future work.

## 2. Background of the FastSLAM

Before introducing the FastSLAM algorithm, we will briefly describe the SLAM problem. SLAM means that the robot determines its trajectory through its own sensors in a location environment and at the same time constructs an environmental map. The robot SLAM problem can be regarded as a kind of probability problem. The robot can achieve the best estimation of pose state and environment map by observing and controlling quantity. Its expression is as follows:(1)$$p(x_{t},m,z_{t},u_{t},n_{t})$$
where $x_{t}$ represents the robot’s position at time t, m represents the map, $z_{t}$ and $u_{t}$ are the measured quantity and the controlled quantity, and $n_{t}$ represents the data association. Meanwhile, the SLAM problem can be represented by a dynamic Bayesian network in Figure 1.

FastSLAM is an excellent algorithm for solving SLAM problems. In FastSLAM, based on the Rao-Blackwellized particle filter, Equation (1) is decomposed into robot path estimation and independent feature estimation. Therefore, the decomposition of Equation (1) can be expressed as:(2)$$p(x_{t},m|z_{t},u_{t},n_{t})=p(x_{t}|z_{t},u_{t},n_{t})\prod_{k=1}^{M}p(m_{k}|x_{t},z_{t},u_{t},n_{t})$$
where $p(x_{t}|z_{t},u_{t},n_{t})$ represents the estimation of the pose of the robot, and $p(m_{t}|x_{t},z_{t},u_{t},n_{t})$ represents the estimation of the environmental features. In the FastSLAM algorithm, the robot pose is estimated by PF, and the environment features are estimated by the extended Kalman filter (EKF). In this process, each particle corresponds to a robot path estimation and environmental features. The particles in FastSLAM can be expressed as:(3)$$X_{k}^{i}=\{x_{k}^{i},u_{1,k}^{i},\sum {}_{1,k}^{i}\cdots u_{N,k}^{i},\sum {}_{N,k}^{i}\}$$
where i is the particle index, $x_{k}^{i}$ is the robot path estimation, $u_{n,k}^{i}$ and $\sum {}_{N,k}^{i}$ is the mean value and covariance of the *n*th feature of the *k*th particle.

The equations above are the key mathematical point of view of the FastSLAM algorithm, for the detailed proof of the FastSLAM algorithm, according to previous literature [31]. Since the proposed distribution function selection method is different, the FastSLAM algorithm is divided into the FastSLAM1.0 algorithm and FastSLAM2.0 algorithm. The FastSLAM1.0 algorithm uses the robot motion model as the particle sampling function, so when the motion control input error is large, the estimated state of the system will be inaccurate. In contrast, in the FastSLAM2.0 algorithm, a complete EKF iterative process is first adopted, in which the latest moment of control input and landmark characteristics measurement values are integrated, and then the posterior estimates of robot pose state are used as the particle sampling function, thus improving the estimation accuracy of the robot.

Although the FastSLAM2.0 algorithm adopted a new importance density function, but it still exists particle degradation problems. Aiming to solve this problem, we introduce the resampling strategy. Although this method can solve the problem of particle weight degradation, it also can cause particle diversity loss problems. Therefore, the key issue to improve the accuracy of robot positioning and mapping is the drop in particle weight and the loss of particle diversity. This article improves on the FastSLAM2.0 algorithm.

For the derivation of the FastSLAM2.0, readers can refer to the literature [16,31]. The main steps and formulas of the FastSLAM2.0 are as follows.
(1)Sampling the pose
(4)$$x_{t}^{i}\sim p(x_{t}|x^{t-1,i},\;z^{t},\;u^{t},\;n^{t})$$(2)EKF updates the observed landmark estimates.(3)Importance weight calculation:
(5)$$w_{t}^{i}=\frac{t\arg et\;distribition}{propsal\;distribition}=\frac{p(x^{t,i}|z^{t},\;u^{t},\;n^{t})}{p(x^{t-1,i}|z^{t-1},\;u^{t-1},\;n^{t-1})p(x_{t}^{i}|x^{t-1,i},\;z^{t},\;u^{t},\;n^{t})}$$(4)Re-sampling.(5)Unknown data associations.(6)Feature management.

## 3. Lion Swarm Optimization Algorithm

Lion Swarm Optimization Algorithm (LSO) is a new method to solve the global optimization problem of the objective function. This method completely simulates the lion’s foraging behavior, migration, and population change to solve the problem [30].

In the LSO algorithm, the pride population is divided into the Lion King, the Lioness, and the Cub. The lion is responsible for guarding the territory and protecting the cubs as well as distributing food. The Lioness hunted and raised the cubs. The cubs, also known as follow lions, feed near the lion when they are hungry. After feeding, they learn to hunt from the Lioness. When he grows up, he will be driven out of his territory and becomes a stray lion. After training, the male lion in the stray lion will challenge the status of the original lion. The LSO algorithm is given in the following steps:

Initialization: Like other intelligent group algorithms, the LSO algorithms realize the initialization of the population number N, dimensional space D, individual position $x_{t}$, and the distribution probability of different individuals within the lion group.

In the literature [30], the different types of individuals in a lion group are set as follows. In a lion group, the number of adult lion individuals affects the effect of the algorithm—the greater the number of young lion individuals, the greater the difference in population numbers. The algorithm detects strong ability. Moreover, the convergence speed of the algorithm is related to the adult lion. In order to maintain the convergence of the algorithm, the proportion of adult lions β is a random number between 0 and 0.5. Similarly, the proportion of cubs is listed as 1−β.

The lion guardians: When the LSO algorithm is used to solve the optimization problem, the location of each food source represents a feasible solution to the optimization problem, and the size of the adaptive value represents the quality of the solution. The maximum moderate value represents the Lion King. The lion retains his privileges in the best food range. Its motion formula is as follows:(6)$$x_{i}^{k+1}=g^{k}(1+\gamma ||P_{i}^{k}-g^{k}||)$$
where $x_{i}^{k+1}$ represents the new position of the Lion King after the movement, $g^{k}$ represents the optimal position in the *k*th generation group, and γ is a random number generated according to a normal distribution N(0,1); $P_{i}^{k}$ is the most historical record of the *i*th lion in the *k*th generation. 

Lioness Hunt: In the process of the Lioness, the two Lioness cooperate to complete the hunt, then the position of the Lioness after cooperative hunting:(7)$$x_{i}^{k+1}=\frac{P_{i}^{k}+P_{c}^{k}}{2}(1+\alpha_{f}\gamma )$$
where $P_{i}^{k}$ is the historical optimal position of the *k*th lion in the i generation. $P_{c}^{k}$ is the historical best position of a hunting partner randomly selected from the k generation lioness group and γ is the random number generated according to the normal distribution N(0,1).

The $\alpha_{f}$ in the above formula represents the disturbance factor of the Lioness’s moving range. The reason for setting the disturbance factor is described in detail in reference [29], and the setting formula is as follows: (8)$$\alpha_{f}=step\cdot \exp{\left(-\frac{30t}{T}\right)}^{10}$$
where ${\bar{g}}^{k}=\bar{low}+\bar{high}-g^{k}$ represents the maximum moving step length in the range of activity, $\bar{high}$ and $\bar{low}$ represents the minimum and maximum lion’s moving values in the range of activity. T is the maximum number of iterations of the population, and t is the number of iterations of the current population.

Lion Cubs Follow: There are three main activities for kindergarten teachers in the lion group. (1) When they are hungry, they will turn to the Lion King; (2) When they are full, they will hunt with the Lioness; (3) When they grow up, they will be driven out of the territory by the Lion King and become A stray lion. The male lion among the lions will challenge the status of the original Lion King. Therefore, the position of the Cub after moving can be represented as:(9)$$x_{i}^{k+1}=\{\begin{array}{c} \frac{g^{k}+P_{i}^{k}}{2}(1+\alpha_{c}\gamma ),q\leq \frac{1}{3}\\ \frac{P_{m}^{i}+P_{i}^{k}}{2}(1+\alpha_{c}\gamma ),\frac{1}{3}\leq q\leq \frac{2}{3}\\ \frac{{\bar{g}}^{k}+P_{i}^{k}}{2}(1+\alpha_{c}\gamma ),\frac{2}{3}\leq q\leq 1 \end{array}$$
where $\gamma ,P^{k},g^{k}$ represent the same meaning as Equations (4) and (5), $P_{m}^{i}$ represents the best position, in the history of the i generation of cubs following lionesses, $\alpha_{c}=step(\frac{T-t}{T})$ represents the movement disturbance factor of the Cub, and ${\bar{g}}^{k}=\bar{low}+\bar{high}-g^{k}$ represents the lion cub being driven away from the Lion King.

Location update: The greedy rule is used to select the individual positions of the lions before and after the update, and the global optimal is updated.

The above description is the main steps of the lion group algorithm. The lion group algorithm sets the Lion King, Lioness, cubs, and simulates the life tyle of the lion group by appropriate values to complete the optimal solution of the problem. Its global convergence and robustness can effectively avoid the premature problem, especially for multi-peak and high-dimensional complex functions.

## 4. Lion Swarm Optimization Algorithm Improves FastSLAM

In the first section, particle mass degradation and particle diversity loss exist in FastSlam2.0 algorithm, where the mobile robot positioning map accuracy caused serious impact. In the literature [17,18,19,20,21,22,23,24,25,26,27,28,29], researchers proposed different improvement strategies. One was to apply the global optimal value algorithm to change the particle set to avoid falling into the local optimal. The common improved method is the combination of intelligent swarm algorithm and the particle filter, which attracts and repels particles through the global optimal value so that particles are concentrated and distributed in the high likelihood region, improving the filtering accuracy, particle degradation and dilution of the algorithm as results. However, this kind of algorithm has the risk of prematurity during algorithm iteration. The other is to improve the particle weight degradation and diversity loss by optimizing or replacing the particle resampling strategy, thereby improving the filtering performance of the algorithm. However, in complex environments and scenes with strong noise, a large number of particles are required to achieve state estimation, which may lead to long running time and low operating efficiency. Inspired by the work in the literature [29], this paper realized the optimization of the FastSLAM algorithm through the improved lion swarm optimization algorithm. The key idea is to optimize the particle distribution after important sampling by the FastSLAM algorithm through LSO, to make the particle distribution after important sampling into a high likelihood region, thus solving the particle weight reduction and improving the positioning accuracy of the robot. 

At the same time, the survival strategies of all kinds of individuals in the lion swarm algorithm were improved and optimized. After the improvement of the Lion King position updating and the Cub following strategy, the optimization ability of particles was further enhanced. The lioness hunting process was optimized by a genetic algorithm, so as to avoid the local optimal situation of the improved algorithm. Furthermore, the degradation of particle weight and loss of particle diversity are proved to improve the positioning accuracy of the robot. The main improvement steps of the algorithm are as follows.

A.Improved Lion position update strategy

In the original LSO algorithm, the Lion King uses Equation (4) to update the position. In this process, the Lion King moves at the optimal moderate value to maintain his privileges. If the original position update formula is directly introduced into the improved algorithm, it will inevitably cause the appropriate value corresponding to the newly generated Lion King position to be lower than the original appropriate value. This will lead to a waste of computing resources and affect the positioning accuracy of the robot. This paper was inspired by literature [32] to reset the new strategy of lion position updating.

First, a set with the number N+1 centered on the current position of the lion is constructed, the set settings are as follows:(10)$$x_{k}^{i}\sim [x_{k}^{i},x_{k}^{i}\pm j\Delta ](j=1,2\cdots \frac{N}{2},\Delta =\bar{high}\cdot 10^{-4})$$
where N represents the number of particles, Δ represents the move step, and $\bar{high}$ represents the distance of the particle with the longest distance from the global optimal value.

Secondly, set the step size threshold $\Delta_{\max}$, judge the moving step size Δ. When $\Delta \geq \Delta_{\max}$, take the moving step size as $\Delta =\Delta_{\max}$, instead, take the current move step as the Lion’s move step. Through this step, the automatic adaptation of the step length is realized. In the early stage of the algorithm, the distance between each particle is relatively large, and the movement step length obtained by calculation must be greater than the threshold. The threshold is used as the current movement step to ensure that the Lion King is performed at a small accurate update range, and when the algorithm enters the later stage, the particle spacing is reduced, and the moving step is smaller than the threshold, and the current moving step is used as the Lion King’s moving step to ensure the Lion King’s update.

Finally, the weight corresponding to each individual in the new particle set is calculated, and the individual with the largest weight is selected as the current new Lion King. Through the new Lion King update strategy, it is ensured that every time the Lion King forages, an individual better than the current Lion King will be generated, and then the particle collection will be moved to the high-likelihood area, thereby improving the filtering accuracy of the algorithm.

B.Reset the Lioness Hunting Method

In the original LSO algorithm, the lioness hunts by two lionesses hunt together, and the two lionesses are in the same position after hunting. When this step is applied to the improvement of the FastSLAM algorithm, in the late running stage of the algorithm, the distance between the particles is relatively close, so adopting this step will further reduce the diversity of particles. In this paper, the crossover step in the genetic algorithm is used as an improved lioness hunting method. The equation is as follows:(11)$${\tilde{x}}_{k}^{m}=ax_{k}^{m}+(1-a)x_{k}^{n}$$
(12)$${\tilde{x}}_{k}^{n}=ax_{k}^{n}+(1-a)x_{k}^{m}$$

In Equations (11) and (12), ${\tilde{x}}_{k}^{m}$ and ${\tilde{x}}_{k}^{n}$ respectively represent the positions of the two lionesses participating in hunting, $x_{k}^{m}$ and $x_{k}^{n}$ represent the positions corresponding to the two lionesses after hunting, and a represents the crossover probability, which is 0.7.

C.Cub Follow Formula Selection

In the Cub follow Equation (7), the cub position update has three different strategies, which are moving to the Lion King, following the Lioness, and staying away from the lion group. In order to improve the filtering accuracy of the robot, in this paper, the position of the Cub is set to be updated as the Cub moves towards the Lion King, and its position updating equation is shown below.
(13)$$x_{i}^{k+1}=\frac{g^{k}+p_{i}^{k}}{2}(1+\alpha_{c}\gamma ),q\leq \frac{1}{3}$$

The parameter setting of Equation (13) is the same as that of Equation (9). By improving following strategy, the particles represented by the young lion are concentrated and distributed in the Gauss natural region in the FastSLAM algorithm after the important sampling of particles is completed. Thus, the positioning and mapping accuracy of the robot is improved.

The improved algorithm flow chart (Figure 2) is shown below:

The above diagram shows the flow chart of the improved algorithm proposed in this paper, in which the blue part is the optimisation of the importance sampling process through the Lioness algorithm, and its detailed process is as follows: firstly, the weights of each particle are calculated, and the composition of the Lioness is designed according to the size of the weights, where the Lioness represents the globally optimal particle, the Lioness consists of particles with good fitness values, and the Cub represents the particle with poor fitness values, and then the optimization of the Lioness is achieved through the three steps above. A. Improved Lion position update strategy, B. Reset the Lioness Hunt-ing Method, C. Cub Follow Formula Selection three steps to achieve the update of the lion population and thus the optimization of the set of particles after importance sampling.

## 5. Performance Analysis

This paper first verifies the feasibility of the algorithm through the MATLAB simulation environment, and then verifies the feasibility of the LSOFastSLAM2.0 algorithm proposed in this paper under the simulated mine environment.

### Simulation

The hardware environment of the experiment is a desktop computer (IntelCorei5 processor, 4 GB memory), and the experiment environment is MATLAB2016b.

First, a mobile robot simulation model is established, in which the mobile robot motion model can be represented as:(14)$$\left[\begin{array}{c} x_{k}^{v}\\ y_{k}^{v}\\ \Phi_{k}^{v} \end{array}\right]=\left[\begin{array}{c} x_{k-1}^{v}+\Delta t\cdot \cos(\Phi_{k-1}^{v}+\theta_{k})\\ y_{k-1}^{v}+\Delta t\cdot \sin(\Phi_{k-1}^{v}+\theta_{k})\\ \Phi_{k-1}^{v}+\frac{\left(\Delta t\cdot v_{k}\sin(\theta_{k})\right)}{D} \end{array}\right]+\left[\begin{array}{c} v_{x}\\ v_{y}\\ v_{\Phi } \end{array}\right]$$

In the formula, $\left(x_{k}^{v},y_{k}^{v}\right)$ represents the position posture state of the robot in the two-dimensional environment at the k moment; $\Phi_{k}^{v}$ represents the heading angle, and the value range is $\left[-180^{{}^{\circ}},180^{{}^{\circ}}\right]$; $v_{k}$ represents the movement speed of the robot, $\alpha_{k}$ represents its steering angle, and $\theta_{k}$ is the robot odometer sampling time, $v_{x},v_{y},v_{\Phi }$ is the noise during the robot movement, D is the distance between the drive shafts.

Then the observation model of the robot is established, and its formula is as follows:(15)$$\left[\begin{array}{c} r_{k}\\ \theta_{k} \end{array}\right]=\left[\begin{array}{c} \sqrt{{\left(x_{i}-x_{k}^{v}\right)}^{2}+{\left(y_{i}-y_{k}^{v}\right)}^{2}}\\ \arctan\frac{y_{i}-y_{k}^{v}}{x_{i}-x_{k}^{v}}-\Phi_{k}^{v} \end{array}\right]+\omega_{k}$$

In the formula, $r_{k},\theta_{k}$ respectively represents the distance between the detected environmental feature and the mobile robot, and the angle of movement direction; $\omega_{k}$ is the observation noise.

The motion parameters and noise parameters of the mobile robot are shown in Table 1 below. The control noise of the robot and the observation noise are three-dimensional. In the following table, the speed noise of the robot represents the noise in the direction of the robot’s movement process, which is composed of *x*-axis noise and *y*-axis noise.

Secondly, simulate the robot localization and mapping, establish the working environment, as shown in Figure 3, set the 17 heading points, 35 road marking points and the mobile robot movement range is 100 m × 80 m. The mobile robot starts from the origin of the coordinates (red point in Figure 3) and moves counterclockwise; the green ∗ represents the road marking point, red ∘ represents the heading point, and cyan line represents the specified robot path.

Finally, in order to verify the effectiveness of the algorithm proposed in this paper, comparative algorithms were also performance which includes: the classic FastSLAM2.0 algorithm, the improved FastSLAM2.0 algorithm based on the gravitational field algorithm (GFA-FastSlAM2.0) [33], and the algorithm proposed in this paper (LSO-FastSLAM2.0). GFA-FastSlAM2.0 has been described in detail in reference [33], but for the sake of completeness, we will introduce its main idea. This algorithm introduces the optimization idea of gravitational field in particle resampling, and the sampled particles are regarded as the cosmic dust system. Each dust receives the action of the movement factor and the rotation factor of the central dust with the largest weight, so as to optimize the distribution of the sampling particles of the mobile robot pose so that the particle set can move towards the real particle more quickly. The pose state of the robot can be approximated to make it converge faster, and the particle degradation and depletion problems that are prone to occur in the FastSLAM2.0 algorithm are improved.

Since the motion noise and observation noise are random, in order to verify the feasibility of the improved algorithm proposed, this paper uses the root mean square error (RMSE) as the criterion. When the number of particles is 20, 50, 80, 100, The FastSLAM2.0 algorithm, GFA-FastSLAM2.0 algorithm, and LSO-FastSLAM2.0 algorithm were performed 20 experiments to obtain the RMSE of the robot position estimation and the road sign estimation, and further, take the average value. The experimental results are shown in Table 2 below.

It can be seen from the above table, the improved algorithm proposed in this paper gradually decreases with the increase of the number of particles, and the positioning accuracy of the algorithm and the accuracy of the road signs gradually decrease. And with the increase of the number of particles, the filtering accuracy of the algorithm tends to stabilize, which is consistent with the convergence characteristics of the FastSLAM algorithm, proving the feasibility of an improved algorithm.

When the number of particles is 20, we verify the improvement degree of the algorithm proposed in this paper about the problem of particle degradation and compares the effective number of particles Neff in the three algorithms. As shown in Figure 4, the effective particle number of the algorithm proposed in this paper is higher than that of the other two algorithms. After calculation, in the process of positioning the three algorithms, the average number of effective particles is 13.8256, 16.3859, and 17.1674, respectively. This phenomenon may owe to the fact that the algorithm proposed in this paper effectively solves the problem of particle weight degradation, and GFA-FastSLAM2.0 algorithm. This phenomenon shows that the algorithm proposed in this paper efficiently optimizes the particle set and solve the problem of particle weight degradation.

In Figure 5, the green and red triangles represent the real and predicted positions of the robot, and the yellow lines represent the observation of the robot on the road markings. It can be seen from Figure 5 that the improved algorithm proposed in this paper has the highest degree of coincidence with the real trajectory, followed by GFA-FastSlam2.0 algorithm, and the FastSlam2.0 algorithm has the worst effect. This indicates that the algorithm proposed in this paper has the highest positioning accuracy among several algorithms. The reason for this phenomenon is that the FastSLAM2.0 algorithm has serious particle degradation and loss of particle diversity in the later stage of the algorithm. The GFA-FastSLAM2.0 algorithm has a higher filtering accuracy, because it acts on the particles through the gravitational field to distribute the particles in the high-likelihood area, which effectively alleviates the problem of particle degradation, and therefore improves the filtering accuracy of the robot. The LSO-FastSLAM2.0 algorithm has the highest positioning accuracy, compared with the optimization of the gravity field algorithm, the Lion algorithm is more effective in particle optimization, so it can effectively solve the problem of particle weight degradation and improve the filtering accuracy of the robot.

In order to further verify the optimization effect of PCSA-FastSlam2.0 on robot positioning and mapping, the predicted and estimated Euclidean distances of the three algorithms were compared at 8000 sampling points. The formula is:(16)$$\rho =\sqrt{{\left(x_{1}-x_{2}\right)}^{2}+{\left(y_{1}-y_{2}\right)}^{2}}$$

In the formula, $\left(x_{1},y_{1}),(x_{2},y_{2}\right)$ represents the coordinates of predicted position and actual position.

The comparison of the robot localization accuracy errors of the three algorithms are shown in Figure 6.

It can be seen from Figure 6 that the algorithm proposed in this paper has the smallest error and is relatively stable, while the positioning accuracy error of the classic FastSLAM2.0 algorithm gradually increases as the positioning accuracy error of the algorithm increases as the running time increases. The reason for this problem is that in the later period of the iterative algorithm, the particles are severely degraded and the diversity of the particles is lost, resulting in lower robot positioning accuracy. The improvement ideas proposed in this article effectively solve this problem and increase the positioning accuracy of the robot. The positioning accuracy of the GFA-FastSLAM2.0 algorithm is lower than that of the improved algorithm proposed in this paper, but it is better than the classic FastSLAM2.0 algorithm. This may be because of the improved particle filter by the gravity field algorithm. The gravity field algorithm has a certain degree of optimization for the particles after importance sampling, and a certain extent to improves the diversity of the particles.

After comparing the average error and variance of the positioning accuracy of the three algorithms, it is obvious that the improved algorithm has improved the robot positioning effect, as shown in Table 3 below.

In the above table, the LSO-FastSLAM2.0 algorithm has the smallest average error and the smallest positioning accuracy variance, indicating that the positioning accuracy and stability of the LSO-FastSLAM2.0 algorithm have been significantly improved. In the GFA-FastSLAM 2.0 algorithm, the strategy for sampling particles is adjusted so that the centers of the sampled particles are distributed near the global optimal value. Due to the unique strategy of simulating gravitational fields, the central dust attracts and repels the surrounding dust and adjusts the distribution of particles. Compared with the FastSLAM algorithm proposed in this paper, GFA-FastSLAM only attracts and repels particles, while LSO-FastSLAM divides particles into three different sets of the Lion King, Lioness, and young lion, and then through different formulas, the optimization of particle sets is realized, and the optimization process is more refined, so the accuracy and stability of the algorithm are improved.

In order to verify the degree of improvement in the accuracy of robot localization and mapping, we compared the RMSE of the *x*-axis, *y*-axis, and road signs, respectively, as shown in Table 4.

It can be seen from Table 4 that the improved algorithm proposed in this paper is better than FastSLAM2.0 and GFA-FastSLAM2.0 algorithms in the *x*-axis, *y*-axis, and road sign estimation. Compared to the FastSLAM2.0 algorithm, the LSO-FastSLAM2.0 algorithm significantly improves the robot positioning and mapping. Compared with GFA-FastSLAM2.0, LSO-FastSLAM2.0 also has a significant improvement, which shows that the algorithm proposed in this paper more effectively optimizes the importance sampling process so as to achieve a significant improvement of the robot positioning map building accuracy.

## 6. Conclusions

In order to improve the accuracy of mine rescue robot positioning and mapping, this paper proposes an improved FastSLAM algorithm based on the lion swarm optimization algorithm. The lion swarm optimization algorithm optimizes the particle distribution after sampling so that the particle set is distributed in the high-likelihood area, thus solving the particle weight degradation and loss of particle diversity in the FastSLAM algorithm thereby improving the accuracy of robot positioning and mapping. The robot simultaneous positioning and mapping experiments show that the improved algorithm not only improves the positioning accuracy of the robot but also improves the algorithm stability. In future work, the improved algorithm will be applied to mine rescue robots to further verify the feasibility of the algorithm.

## Figures and Tables

**Figure 1 sensors-22-01297-f001:**
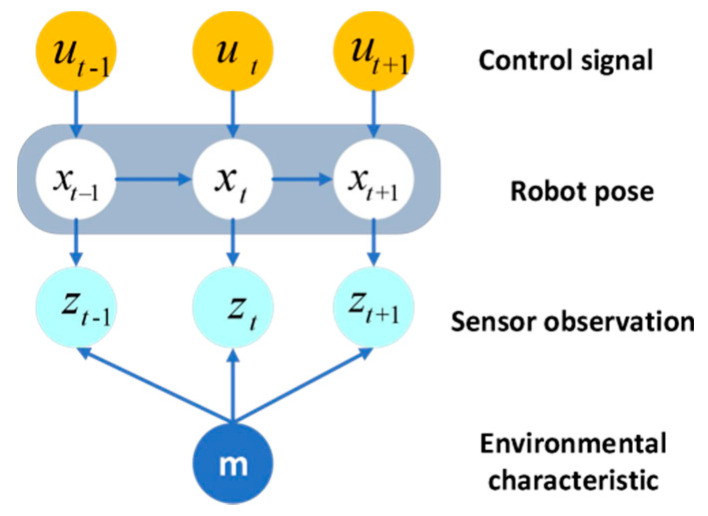
Dynamic Bayesian networks for SLAM.

**Figure 2 sensors-22-01297-f002:**
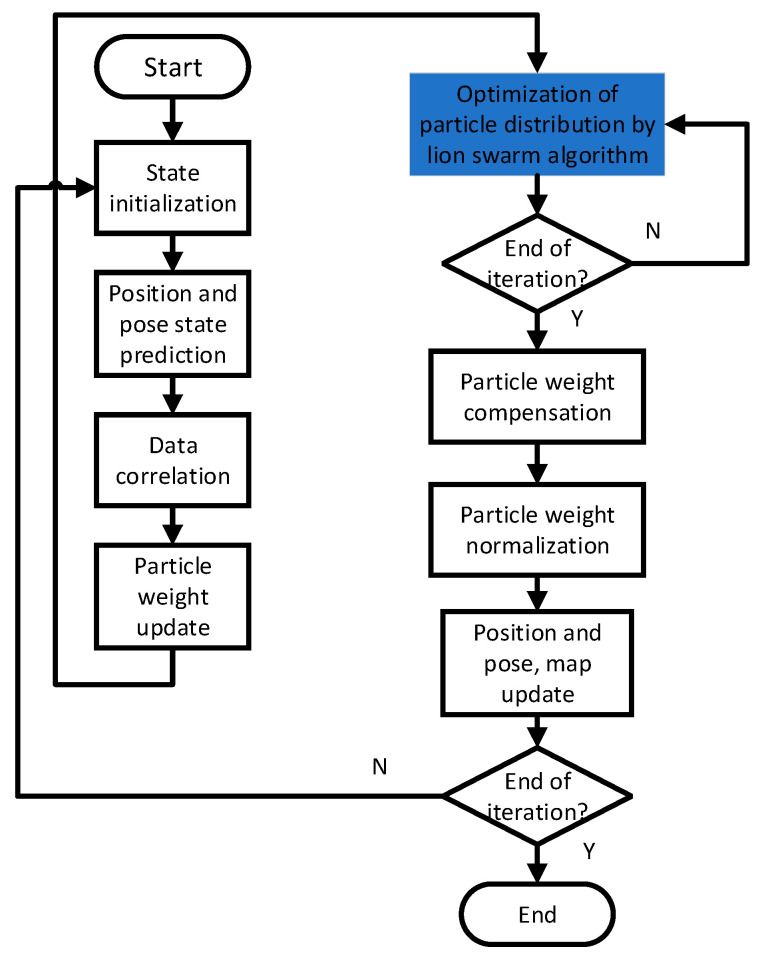
Flowchart of improved algorithm.

**Figure 3 sensors-22-01297-f003:**
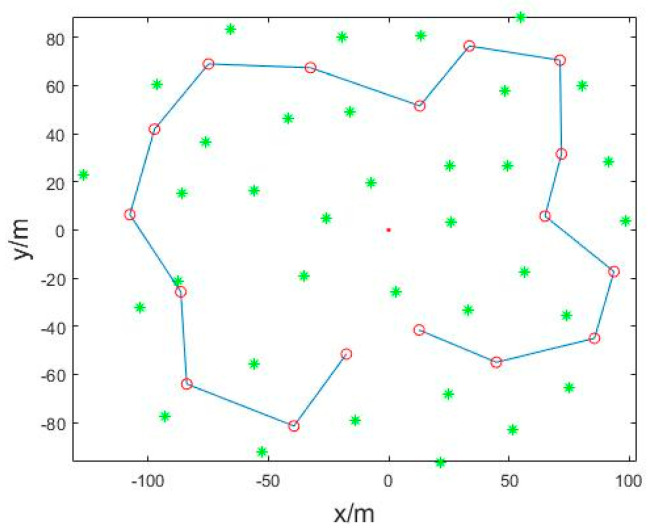
Simulation Environment.

**Figure 4 sensors-22-01297-f004:**
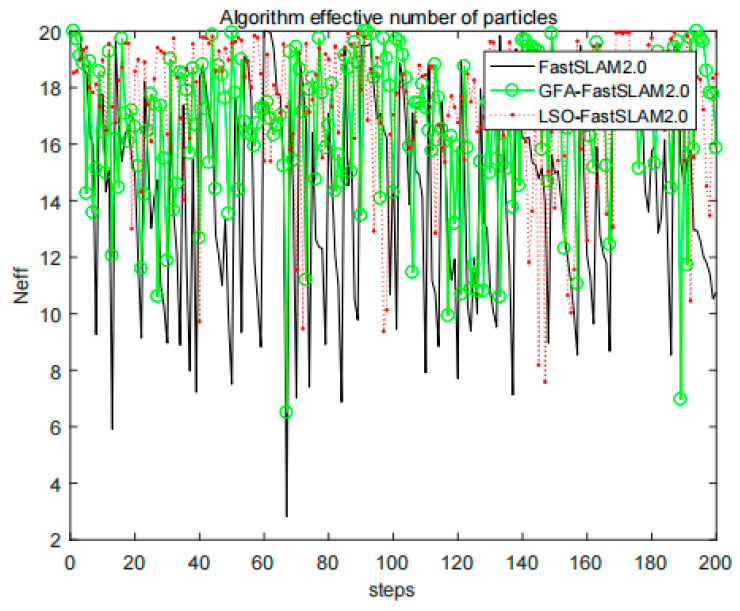
Effective particle number comparison.

**Figure 5 sensors-22-01297-f005:**
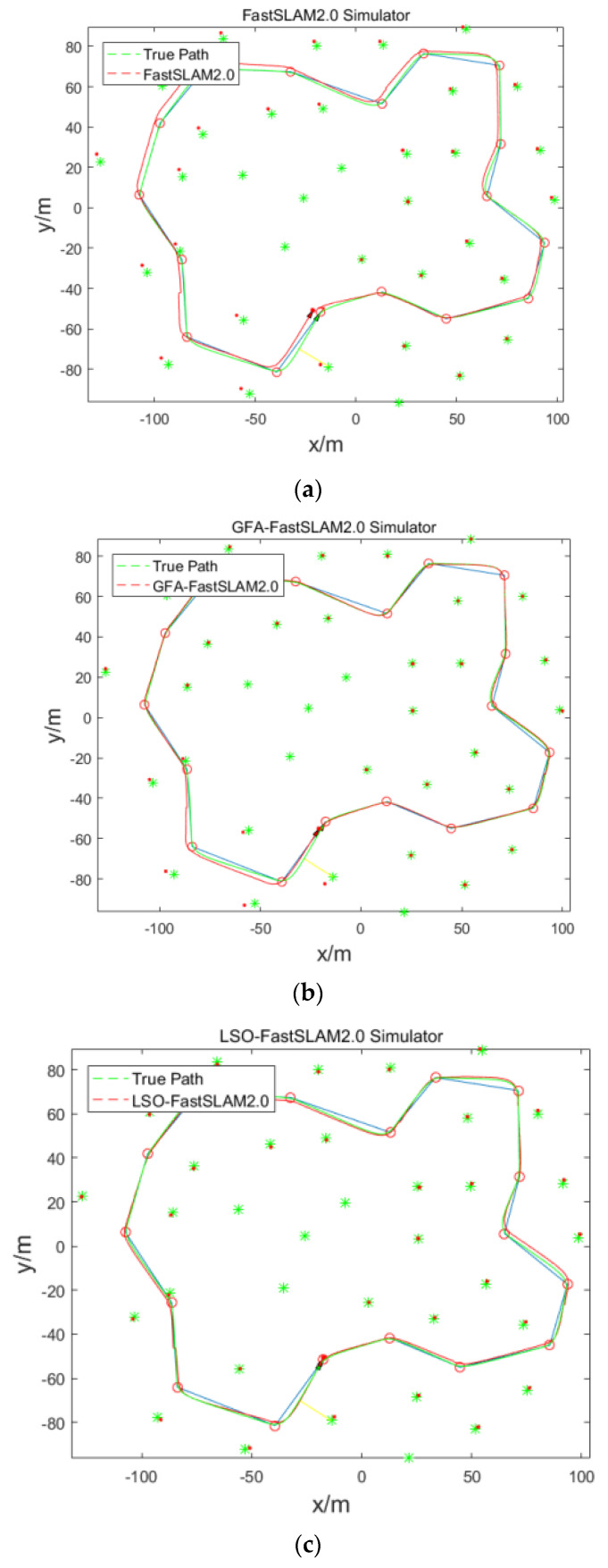
The algorithm real trajectory: (**a**) FastSLAM2.0 Simulation Result; (**b**) GFA-FastSLAM2.0 Simulation Result; (**c**) LSO-FastSLAM2.0 Simulation Result.

**Figure 6 sensors-22-01297-f006:**
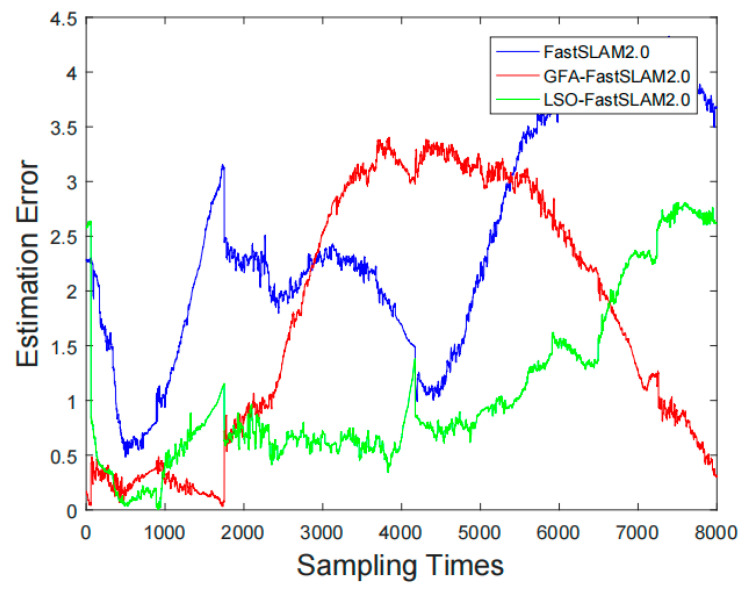
Comparison Chart of Robot Localization Accuracy Error.

**Table 1 sensors-22-01297-t001:** Motion parameters and noise parameters of mobile robots.

Parameter	Numerical Value	Noise Parameters	Numerical Value
Robot speed	3 m/s	Motion noise	0.3 m/s 1.5°
Max steering angle	10°		
Maxi steering angular speed	15°/s	Observation noise	0.1 m/s 1°
Wheel spacing	4 m		
Sampling time interval	0.025 s		

**Table 2 sensors-22-01297-t002:** Improved algorithm validity proof.

Number of Particles	Algorithm	Mean Localization Accuracy Error/M	RMSE of Road Sign Estimation (m)
20	FastSLAM2.0	3.0535	4.1399
	GFA-FastSLAM2.0	2.9060	3.3545
	LSO-FastSLAM2.0	2.3025	2.2837
50	FastSLAM2.0	2.7718	3.2106
	GFA-FastSLAM2.0	2.3629	2.5990
	LSO-FastSLAM2.0	2.0470	2.2837
80	FastSLAM2.0	2.6843	2.9199
	GFA-FastSLAM2.0	1.7504	1.9072
	LSO-FastSLAM2.0	1.2745	1.3762
100	FastSLAM2.0	2.5907	2.8538
	GFA-FastSLAM2.0	1.3693	1.6422
	LSO-FastSLAM2.0	1.1745	1.3279

**Table 3 sensors-22-01297-t003:** Comparison of mean error and variance of localization accuracy of three algorithms.

Algorithm	Mean Localization Accuracy Error/m	Variance of Localization Accuracy Error
FastSLAM2.0	2.7718	1.6403
GFA-FastSLAM2.0	1.4036	0.9059
LSO-FastSLAM2.0	1.1867	0.2519

**Table 4 sensors-22-01297-t004:** Comparison of RMSE of three algorithms: *x*-axis, *y*-axis and road sign estimation.

Algorithm	RMSE of *x*-Axis (m)	RMSE of *y*-Axis (m)	RMSE of Road Sign Estimation (m)
FastSLAM2.0	2.0447	2.2676	2.9871
GFA-FastSLAM2.0	1.6015	1.1018	1.5841
LSO-FastSLAM2.0	0.6932	1.0518	1.3383

## Data Availability

Not applicable.

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
