# Peer review of "LSO-FastSLAM: A New Algorithm to Improve the Accuracy of Localization and Mapping for Rescue Robots"

_sensors, 2022, doi:10.3390/s22031297_

Round 1

Reviewer 1 Report

General Comments:

This manuscript aims to improve the accuracy of mine robot positioning and mapping for a rapid rescue. The selected FastSLAM algorithm is improved with the lion swarm optimization method. The objective and results of the study are very interesting; however, the manuscript has several limitations, in the following specific sections.

 Specific comments:

Title: Please modify the Title and highlight the aim of the study in more precise manner. It sounds quite simple.

 Abstract:

  • Please briefly describe the solution and reason of selection of specific algorithm for the mentioned problem.
  • Please avoid the extra space between the words in the same sentence.

 Introduction:

  • Provide some literature data about some previous studies which utilizes the specific algorithm.
  • Please define the problem and the solution in a consistent manner.
  • Add literature data about the optimization approaches in order to improve the accuracy in general and for the specific algorithm.
  • What about adding value of your paper? Please define.
  • In introduction section, page 3 first paragraph has two full stops.
  • Only accuracy improvement is not enough novelty to get it accepted. Please make a separate section for your contributions and describe in at least 3-4 folds.
  • Also summarize the previous work through a comparison table. That should highlight the pros and cons of the previous methods as well as proposed method.

 Materials and methods:

  • Please describe the optimization approach in a more consistent and structured manner.
  • Please describe the GFA-FastSLAM2.0 in more detailed.
  • Please be more consistent in the description your approach in order to differentiate with the previous ones, these are too reductive.
  • How the problem of particle degradation is overcome by the LSO-FastSLAM2.0? Please specify.
  • GFA-FastSLAM2.0 has low accuracy error than the proposed approach? Please define it in a clear way?
  • Please rewrite the last section of “Performance analysis” which including the Table 3 and Table 4 in a more structured manner.

Results:

  • Please rewrite the last section of “Performance analysis” which including the Table 3 and Table 4 in a more structured manner. it seems too confusing and difficult to follow.

 Tables:

  • Please improve the layout.

Figures:

  • Please rewrite almost all figure captions in a more synthetic manner.

Linguistic and typewriting: English writing needs some improvements.

Reviewer 2 Report

This paper proposes a FastSLAM algorithm based on the lion swarm optimization method. The experimental results show that the proposed method outperforms the state-of-the-art approaches.

There are some problems of theoretical and experimental analyses in this manuscript and it can be revised in the following aspects.

  1. The paper title is a sentence. It is not a suitable choice.
  2. The whole article should be carefully checked to avoid some possible typographical and grammatical errors. For instance, in section 2, “Its expression is as follows” should be “Its expression is as follow”.
  3. All mathematical expressions should be meticulously examined to avert some potential mistakes. For example, under equation (1), “nt” should be “n_t”. For another example, in equation (2), the multiplication operator should be summation operation.
  4. In Tab. 3, the stability rate cannot be close to 100%. High stability rate is very important for effective rescue. How to improve the proposed method?

Round 2

Reviewer 1 Report

Most of my comments are addressed. I recommend acceptance of the paper in current form. 

Reviewer 2 Report

The authors have correctly answered all my questions, the paper has been satisfactorily improved, and it can be considered for further processing.
